# Methods for Obtaining One Single Larmor Frequency, Either *v*_1_ or *v*_2_, in the Coherent Spin Dynamics of Colloidal Quantum Dots

**DOI:** 10.3390/nano13132006

**Published:** 2023-07-05

**Authors:** Meizhen Jiang, Yuanyuan Zhang, Rongrong Hu, Yumeng Men, Lin Cheng, Pan Liang, Tianqing Jia, Zhenrong Sun, Donghai Feng

**Affiliations:** 1State Key Laboratory of Precision Spectroscopy, East China Normal University, Shanghai 200241, China; 52200920010@stu.ecnu.edu.cn (M.J.); 52170920029@stu.ecnu.edu.cn (Y.Z.); 52250920010@stu.ecnu.edu.cn (Y.M.); 51210920033@stu.ecnu.edu.cn (L.C.); tqjia@phy.ecnu.edu.cn (T.J.); zrsun@phy.ecnu.edu.cn (Z.S.); 2College of Sciences, Shanghai Institute of Technology, Shanghai 201418, China; 3College of Arts and Sciences, Shanghai Dianji University, Shanghai 201306, China; liangp@sdju.edu.cn; 4Collaborative Innovation Center of Extreme Optics, Shanxi University, Taiyuan 030006, China

**Keywords:** colloidal quantum dots, spin dynamics, time-resolved spectroscopy, hole acceptors

## Abstract

The coexistence of two spin components with different Larmor frequencies in colloidal CdSe and CdS quantum dots (QDs) leads to the entanglement of spin signals, complicating the analysis of dynamic processes and hampering practical applications. Here, we explored several methods, including varying the types of hole acceptors, air or anaerobic atmosphere and laser repetition rates, in order to facilitate the obtention of one single Larmor frequency in the coherent spin dynamics using time-resolved ellipticity spectroscopy at room temperature. In an air or nitrogen atmosphere, manipulating the photocharging processes by applying different types of hole acceptors, e.g., Li[Et_3_BH] and 1-octanethiol (OT), can lead to pure spin components with one single Larmor frequency. For as-grown QDs, low laser repetition rates favor the generation of the higher Larmor frequency spin component individually, while the lower Larmor frequency spin component can be enhanced by increasing the laser repetition rates. We hope that the explored methods can inspire further investigations of spin dynamics and related photophysical processes in colloidal nanostructures.

## 1. Introduction

The electron spin in semiconductors is a promising candidate for quantum bits which are the elementary units for quantum information processing [1,2,3]. It is of fundamental importance to understand electron spin properties and further control the spins. Compared with bulk semiconductors, the confined electrons in quantum dots (QDs) are more adaptable to quantum information science, due to the less efficient spin–orbit interaction and longer spin dephasing times [4]. Over the past two decades, many efforts have been dedicated to investigating the spin properties and the spin manipulation in colloidal QDs grown using wet chemical approaches, obtaining long spin dephasing times up to few nanoseconds even at room temperature [5,6,7]. The open surface and surrounding matrix of colloidal QDs provide convenience for optical and chemical manipulation of electron spin [8,9,10,11,12]. In addition, colloidal QDs have the advantages of low cost and easily controlled shapes, sizes and structures. Since it was first reported by J. A. Gupta et al. that ensembles of colloidal CdSe QDs show two distinct spin components in the coherent dynamics, which differ in *g* factors and Larmor frequencies [13], colloidal CdS QDs [10] were also found to have similar phenomena. The coexistence of two spin components with different Larmor frequencies (referred to below as *v*_1_ and *v*_2_) entangles the spin signals and complicates the investigation of their origins and the related spin relaxation mechanisms. For instance, in ensembles of colloidal CdSe QDs, the *v*_1_ and *v*_2_ spins with spin dephasing times up to hundreds of picoseconds were assigned to the electron and exciton, respectively (electron–exciton model) [5], which is, however, contradictory to the fact that exciton spin relaxation times are short and have around a subpicosecond time scale [14]. Recently, the two spin components were redemonstrated to both belong to electrons in photocharged QDs, which have different wave function spreads, which were either confined by the QD potential (corresponding to the *v*_1_ component) or additionally localized in the surface vicinity (corresponding to the *v*_2_ component) [11]. Similarly, for CdSe nanocrystals embedded in a glass matrix, a solely detected Larmor precession frequency (corresponding to the *v*_2_ component in solution-grown colloidal CdSe QDs) was also determined to arise from resident electrons localized in the vicinity of the surface [15]. However, a systematic investigation of separation and manipulation of the two spin components is still lacking. Obtaining one pure spin component is not only important for deeper understanding of the underlining spin relaxation mechanism and related photophysics but also crucial for the practical applications of quantum bits.

In this work, we investigated the spin signals of colloidal CdSe and CdS QDs in toluene using time-resolved ellipticity spectroscopy under different photocharging conditions, such as addition of different types of hole acceptors, changing laser repetitions as well as air and nitrogen (N_2_) atmospheres. We systematically studied the methods that can contribute to obtaining or enhancing one single Larmor frequency, either *v*_1_ or *v*_2_, in the coherent spin dynamics. In a N_2_ atmosphere, the addition of hole acceptor, either Li[Et_3_BH] or 1-octanethiol (OT), is beneficial to obtain the spin signal of the *v*_1_ component, while adding the OT hole acceptor in an air atmosphere usually brings the *v*_2_ spin component into being separately. High laser repetition rates can enhance the spin signal of the *v*_1_ component, whereas low laser repetition rates are convenient for obtaining the *v*_2_ component alone.

## 2. Materials and Methods

### 2.1. Sample Preparation

Octadecylamine-stabilized colloidal CdSe (average diameter: 6.5 nm, estimated from the first exciton absorption peak [16]) and oleic-acid-stabilized CdS QDs (average diameter: 5.3 nm) in toluene solution were commercially obtained from Hangzhou Najing Technology Co., Ltd. (Hangzhou, China) The corresponding absorption spectra are shown in Figure 1. The mass concentration of all obtained QD samples is 5 mg/mL. For photocharging experiments, the well-known OT hole acceptor [17] or Li[Et_3_BH] [18], purchased from Sigma Aldrich, was mixed with toluene solutions of CdSe or CdS QDs in an airtight quartz cuvette with a thickness of 1 mm. For Li[Et_3_BH], the sample preparation was performed in a glove box in a N_2_ atmosphere. The OT hole acceptor was mixed with QDs in either a N_2_ or air atmosphere. The final concentrations of all samples in the measurements were kept the same at 2.5 mg/mL.

### 2.2. Setup for the Measurements of Coherent Electron Spin Dynamics

The coherent spin dynamics of colloidal QDs were measured using time-resolved ellipticity spectroscopy [11,19,20]. The experimental configuration depicted in Figure 2 is based on a ytterbium-doped potassium gadolinium tungstate (Yb-KGW) regenerative amplifier (PHAROS, Light Conversion Ltd; the central wavelength is 1026 nm, and the pulse duration is ~270 fs), combined with an optical parametric amplifier (OPA). The laser repetition rates of the OPA outputs are tunable and up to 50 kHz. All the measurements were performed at room temperature and in a transverse external magnetic field provided by an electromagnet with iron poles.

In the pump−probe measurements as shown in Figure 2, the pump and probe pulses are wavelength-degenerate and emitted from the OPA with their wavelength tuned at the low energy side of the first exciton absorption band. The circularly polarized pump pulses generate the spin polarization in the QD samples, while the subsequent dynamics of this spin polarization are monitored by the change of ellipticity of the linearly polarized probe pulses. The linearly polarized probe pulses become partially elliptically polarized after transmitting through the spin-polarized QDs because of the absorption difference of left- and right-circularly polarized light. The time delay between the pump and probe pulses is adjusted by a mechanical delay line.

## 3. Results

### 3.1. Spin Dynamics of As-Grown CdSe and CdS Colloidal QDs with Native Ligands

As-grown colloidal CdSe and CdS QDs with native ligands normally have two spin Larmor frequencies in coherent spin dynamics measured at high laser repetition rates. As shown in Figure 3, time-resolved ellipticity signals of as-grown colloidal CdSe and CdS QDs are measured in a transverse magnetic field *B* of 500 and 300 mT, respectively. The corresponding fast Fourier transform (FFT) spectra are shown in the right panels of Figure 3. Figure 3c is the fast Fourier transform (FFT) spectrum of the spin dynamics of CdSe QDs, which shows two Larmor precession frequencies of *v*_1_ = 7.32 GHz and *v*_2_ = 11.11 GHz. Two *g* factor values of *g*_1_ = 1.05 and *g*_2_ = 1.59 are correspondingly derived from these frequencies by the equation g=hvL/(μBB), where *h*, vL and μB are the Planck constant, Larmor precession frequency and Bohr magneton, respectively. Both *g* factor values are in line with the size dependence of *g* factors reported in the literature [11]. Similarly, Figure 3d shows two Larmor precession frequencies and the corresponding *g* factors of CdS QDs of *v*_1_ = 7.73 GHz, *g*_1_ = 1.85, and *v*_2_ = 8.12 GHz, *g*_2_ = 1.93. Despite the FFT amplitudes and spin dynamic curves of CdSe and CdS QDs being distinctly different, both the spin signals comprise two Larmor frequencies (the smaller one refers to the *v*_1_ component, and the larger one is the *v*_2_ component). The FFT amplitude of the *v*_1_ component is much smaller than that of the *v*_2_ component in CdSe QDs, while it is the opposite in CdS QDs. The difference of relative FFT amplitudes may result from different surface ligands and their interactions with the QDs. It has been shown in the literature that the *v*_1_ component is stronger than the *v*_2_ component for as-grown 6 nm CdSe QDs with stearic acid stabilizing ligands [21], while the *v*_1_ component is significantly weaker than the *v*_2_ component for 6.1 nm CdSe QDs with octadecylamine stabilizing ligands [22]. In the following, we will provide appropriate methods which selectively bring one of the two spin components into being, or purposefully enhance one of them and suppress the other.

### 3.2. Methods to Individually Obtain the v_1_ Component

Adding hole acceptors to the QD solutions is helpful in manipulating the spin signals. Figure 4a,b show time-resolved ellipticity signals in as-grown colloidal CdSe and CdS QDs and QDs with Li[Et_3_BH] hole acceptors prepared in a N_2_ atmosphere. The presence of the hole acceptor Li[Et_3_BH] strongly increases the ellipticity signals both for CdSe and CdS QDs. As depicted in the insets of Figure 4a,b, actually only the *v*_1_ component is remarkably enhanced, while the *v*_2_ component disappears. This indicates that addition of Li[Et_3_BH] hole acceptors in a N_2_ atmosphere is conducive to separating the entangled spin signals by increasing the *v*_1_ component and suppressing the *v*_2_ component. The reason is that the Li[Et_3_BH] hole acceptor, as a strong chemical reducing agent, can capture the photogenerated holes and leave the electrons in the dot core (corresponding to photocharging state of the *v*_1_ component [11]). With enough Li[Et_3_BH] molecules, all QDs could be charged to this state, and the spin signal of the *v*_2_ component (which corresponds to another charging state [11]) is thus suppressed. With the removal of the influence of oxygen (as electron scavengers [23,24]) in a N_2_ atmosphere, the photodoped electrons in the dot core are long-lived [18]. Under the irradiation of ambient room light and pump/probe pulses, the spin amplitudes of CdSe and CdS QDs with Li[Et_3_BH] remain almost the same after two hours as shown in Figure 4c, indicating that stable photocharging can be obtained in CdSe and CdS QDs. Stable charging means that the generation of photocharging states can adequately compensate for their decay, which requires appropriate Li[Et_3_BH] concentrations and appropriate light illumination conditions.

Consequently, an effective strategy to achieve pure states of the *v*_1_ component is to modify colloidal QDs with Li[Et_3_BH] hole acceptors in an anaerobic atmosphere. The spin signals of the *v*_1_ component are considerably amplified, which also improves the signal-to-noise ratio significantly.

### 3.3. Methods to Individually Obtain the v_2_ Component

Inspired by the success of individually obtainment of the *v*_1_ spin component by modifying QDs surface with an appropriate hole acceptor in a N_2_ atmosphere, we investigated the effect of adding another OT hole acceptor to QDs prepared in an air atmosphere. The comparison of the time-resolved ellipticity signals of CdSe QDs with and without OT prepared in an air atmosphere is shown in Figure 5a. Adding OT, the spin signal of the *v*_2_ component increases to about 8 times that of as-grown QDs. The inset of Figure 5a shows that the FFT amplitude of *v*_2_ enlarges, while *v*_1_ disappears. A similar phenomenon can be found in the measurements in as-grown CdS QDs and QDs with OT prepared in an air atmosphere, as depicted in Figure 5b. The spin amplitude of the *v*_2_ component significantly increases and brings the *v*_2_ component into being separately. Note that the laser repetition for CdS QDs with OT is 10 kHz. Since with the higher laser repetition, i.e., with 50 kHz laser repetition, the spin signal of the *v*_1_ component is apparent as a result of the pile-up effect from previous pulses. The *v*_1_ component can appear at high laser repetition rates, and the higher the laser repetition rate, the stronger the intensity of the *v*_1_ component, which will be illustrated below. For measurements with different laser repetition rates, a normalization of both pulse energies and laser powers is requisite. The spin signal of CdS QDs with OT measured at 10 kHz laser repetition rates in the main panel has already been multiplied by a factor of 5; equivalently, both the pulse energy and laser power are normalized to be the same between the measurements with 10 and 50 kHz laser repetition rates.

The following are possible reasons that account for the selective enhancement of the *v*_2_ spin component by using OT hole acceptors. OT molecules are linked to the QD surface by replacing native ligands and capture the photogenerated holes in QDs. Therefore, the captured holes are still in nearby the QD surface. Due to the Coulombic attraction, the electron also tends to stay at nearby the QD surface. This charging state contributes to the spin signal of the *v*_2_ component [11] and can be enhanced by addition of enough OT molecules. Meanwhile, the charging state corresponding to the *v*_1_ component is totally inhibited due to the presence of oxygen. Those results suggest that addition of OT with an appropriate molar ratio to QDs in an air atmosphere may provide an approach to obtaining the spin signal of the *v*_2_ component solely.

Note that modifying the QDs surface with an OT hole acceptor prepared in a N_2_ atmosphere can also provide an effective strategy to boost the spin signal of the *v*_1_ component. The spin dynamics of CdSe QDs with and without an OT hole acceptor prepared in a N_2_ atmosphere are shown in Figure 6a. Via adding OT, the total spin amplitudes of CdSe QDs are considerably enhanced. The FFT amplitudes of the *v*_1_ component are relatively weak in both as-grown CdSe QDs and QDs with OT (OT/QDs = 7000) compared with those of the *v*_2_ component as shown in Figure 6b. However, increasing the molar ratio of OT to QDs to 70,000, the FFT amplitude of the *v*_1_ component is increased significantly, while the *v*_2_ component is reduced. For CdS QDs, some similar phenomena also occur, e.g., addition of OT enhances the total spin signal, but some details are different. Figure 6c,d show the spin signals of colloidal CdS QDs and QDs with OT prepared in a N_2_ atmosphere and the corresponding FFT spectra. When the molar ratio of OT to QDs is 100, there is only one frequency of the *v*_1_ component, while the *v*_2_ component disappears. However, increasing the molar ratio of OT to QDs to 5000, the *v*_2_ component emerges again. The anaerobic conditions also help the enhancements of the *v*_1_ component with addition of OT hole acceptors in both CdSe and CdS QDs. In some cases, with appropriate OT concentrations, even a pure *v*_1_ spin component could be achieved.

### 3.4. Influence of Laser Repetition Rates on Spin Signals

Figure 7 shows the spin dynamics of as-grown CdSe and CdS QDs (with native ligands at an air atmosphere) at different laser repetition rates and the corresponding FFT spectra. At a high laser repetition rate of 50 kHz, spin dynamics of colloidal QDs, either CdSe or CdS QDs, exhibit two distinct spin components with different Larmor precession frequencies, which is similar to the results shown in Figure 3. As the laser repetition rate decreases from 50 to 1 kHz, the spin amplitude reduces slightly, and the *v*_1_ component gradually disappears in CdSe QDs as shown in Figure 7a,b. At the laser repletion rate of 1 kHz, only the *v*_2_ component is shown in the coherent spin dynamics. The amplitudes of the *v*_2_ component show almost no change when altering the laser repetition rates. As-grown CdS QDs exhibited similar experimental phenomena when reducing laser repetition rates from 50 to 0.5 kHz. Decreasing the laser repetition rates can also apparently decrease the spin signals of the *v*_1_ component but have negligible effects on the *v*_2_ component. This suggests that the charging state of the *v*_1_ component is long-lived and subject to pile-up effects from previous pulses at high laser repetition rates. Decreasing the laser repetition rates decreases the pile-up effects. When pile-up effects are equal to or less important than the oxygen-induced electron removing, the *v*_1_ component disappears. The results in Figure 7 also imply that the charging state of the *v*_2_ component is short-lived and has no pile-up effects from previous pulses. The influence of laser repetition rates demonstrated here explains the fact that, in the literature, only one single Larmor frequency (corresponding to *v*_2_) was observed for pump–probe measurements with low laser repetition rates [7,25,26], while two Larmor frequencies were typically detected for the measurements with high laser repetition rates [5,13,22]. Accordingly, decreasing laser repetition rates is an effective way to isolate the spin signal of the *v*_2_ component. If a stronger spin signal of the *v*_1_ component is required, then high laser repetition rates must be used for the measurements.

For convenience, we summarize the methods that facilitate the acquisition of one single Larmor frequency, either the *v*_1_ or *v*_2_ component, in spin coherent measurements of our samples as follows:(1)Addition of an appropriate molar ratio of hole acceptors, Li[Et_3_BH] or OT, in a N_2_ atmosphere is beneficial to obtain the pure spin signal of the *v*_1_ component.(2)Modifying colloidal QDs with an OT hole acceptor prepared in an air atmosphere provides advantages on the selection of the *v*_2_ spin component.(3)Low laser repetition rates are convenient for the appearance of merely the *v*_2_ component, whereas the spin signals of the *v*_1_ component rise effectively when applying high laser repetition rates.

## 4. Conclusions

In conclusion, the coherent spin dynamics of CdSe and CdS colloidal QDs were studied using time-resolved ellipticity spectroscopy. Two spin components with different Larmor frequencies typically coexist in QDs with native ligands when measured at high laser repetition rates. In the presence of appropriate hole acceptors, e.g., Li[Et_3_BH] and OT, with a proper molar ratio in N_2_ or air atmosphere, the two spin components can be selectively enhanced or suppressed, leaving one pure spin component. Low laser repetition rates are beneficial to the appearance of the *v*_2_ component solely. Higher laser repetition rates significantly contribute to the enhancement of the *v*_1_ spin component. Consequently, those results imply that addition of hole acceptors either in an air or anaerobic atmosphere and adjusting laser repetition rates provide effective strategies to manipulate the spin signals in colloidal QDs, which is helpful for in-depth investigation of spin dynamics and practical applications.

## Figures and Tables

**Figure 1 nanomaterials-13-02006-f001:**
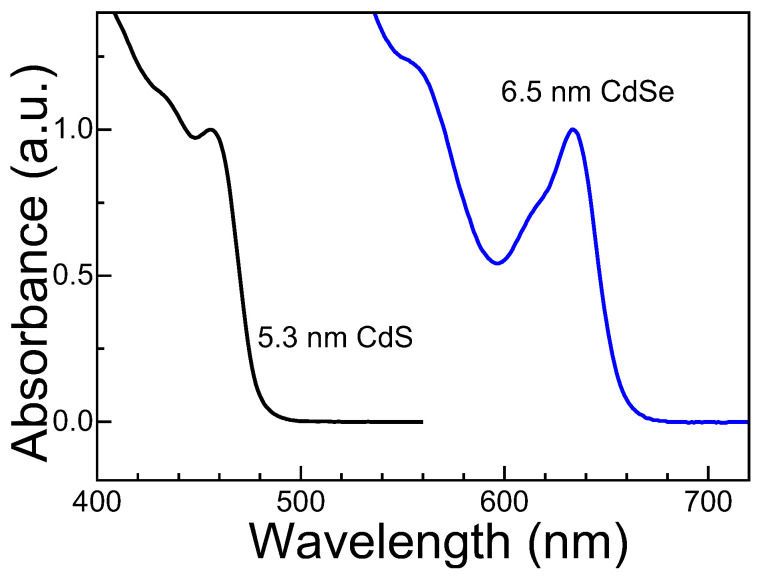
Absorption spectra of as-grown colloidal 6.5 nm CdSe and 5.3 nm CdS QDs. The intensity of the first absorption peak of CdSe and CdS QDs is normalized.

**Figure 2 nanomaterials-13-02006-f002:**
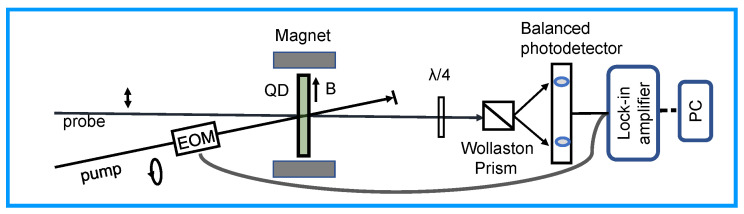
Setup schematic of time-resolved ellipticity measurements.

**Figure 3 nanomaterials-13-02006-f003:**
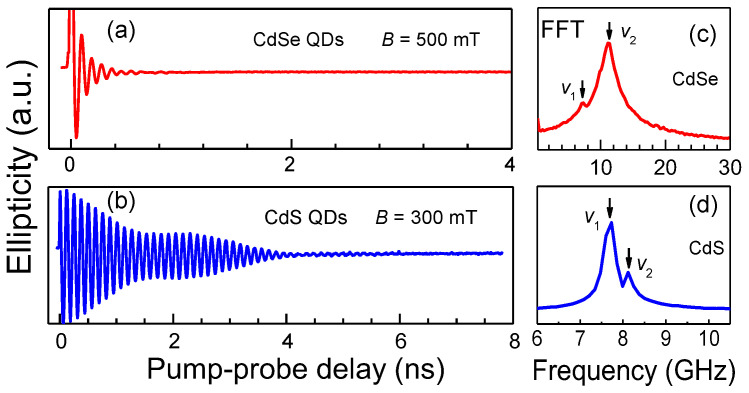
Time-resolved ellipticity signals in as-grown colloidal (**a**) CdSe and (**b**) CdS QDs. (**c**) and (**d**) are the fast Fourier transform (FFT) spectra of panel (**a**) and (**b**), respectively. The laser repetition rate is 50 kHz.

**Figure 4 nanomaterials-13-02006-f004:**
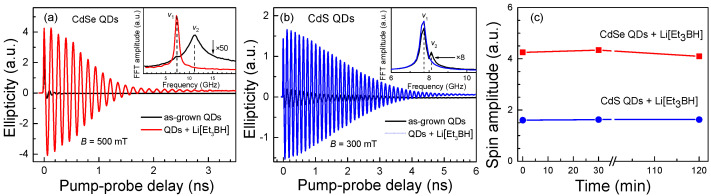
(**a**,**b**) Time-resolved ellipticity signals in as-grown colloidal CdSe and CdS QDs and QDs with Li[Et_3_BH] prepared in a N_2_ atmosphere. The insets are the corresponding FFT spectra of panel a and b. The FFT spectra of as-grown CdSe and CdS QDs are multiplied by 50 and 8, respectively, for better distinction of the two observed Larmor frequencies. (**c**) Spin amplitudes in CdSe and CdS QDs with Li[Et_3_BH] as a function of time under ambient room light illumination. The molar ratio of Li[Et_3_BH] to CdSe and CdS QDs is 500 and 10, respectively. The laser repetition rate is 50 kHz.

**Figure 5 nanomaterials-13-02006-f005:**
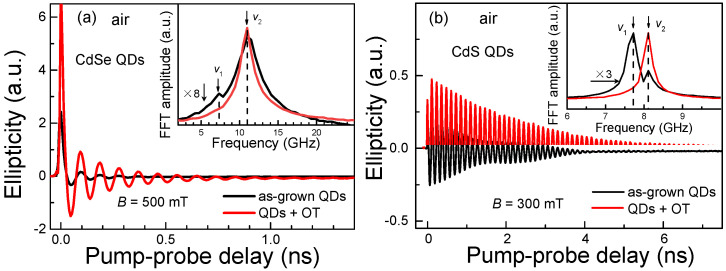
(**a**) Time-resolved ellipticity signals of as-grown colloidal CdSe QDs and QDs with 1-octanethiol (OT) prepared in an air atmosphere. The inset is the corresponding FFT spectra. The FFT spectroscopy of as-grown CdSe QDs is multiplied by 8 for clarity. The molar ratio of OT to CdSe QDs is 11,000. The laser repetition rate is 50 kHz. (**b**) Spin signals of as-grown colloidal CdS QDs and QDs with OT prepared in an air atmosphere. The inset is the corresponding FFT spectra. The FFT spectroscopy of as-grown CdS QDs is multiplied by 7 for clarity. The molar ratio of OT to CdS QDs is 5000. The laser repetition rates for as-grown CdS QDs and QDs with OT are 50 and 10 kHz, respectively.

**Figure 6 nanomaterials-13-02006-f006:**
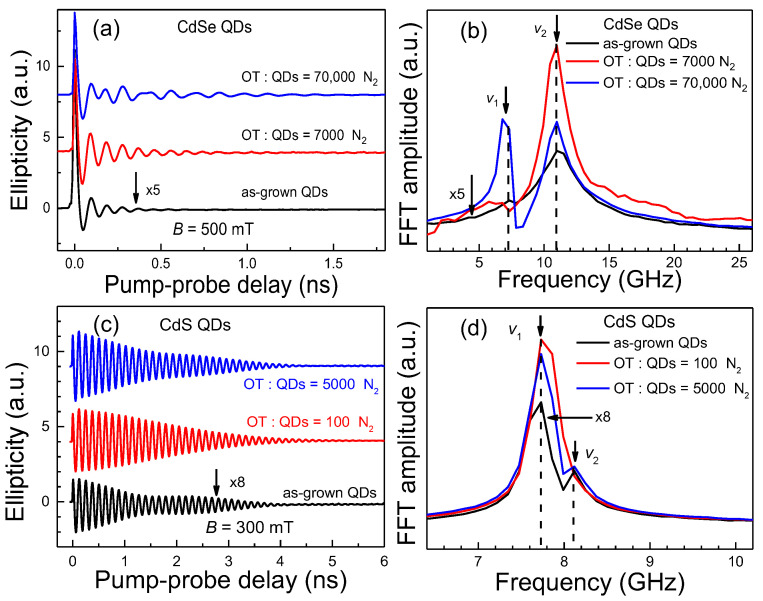
(**a**) Time-resolved ellipticity signals in as-grown colloidal CdSe QDs and QDs with OT prepared in a N_2_ atmosphere. (**b**) The corresponding FFT spectra of panel a. The spin signal and its FFT spectroscopy of as-grown CdSe QDs are both multiplied by 5 for better clarity. The laser repetition rate is 50 kHz. (**c**) Spin signals of colloidal CdS QDs and QDs with OT prepared in a N_2_ atmosphere. (**d**) The corresponding FFT spectra of panel c. The spin signal and its FFT spectroscopy of as-grown CdS QDs are both multiplied by 8. The laser repetition rate is 50 kHz. The plots in panel a and c are offset for clarity.

**Figure 7 nanomaterials-13-02006-f007:**
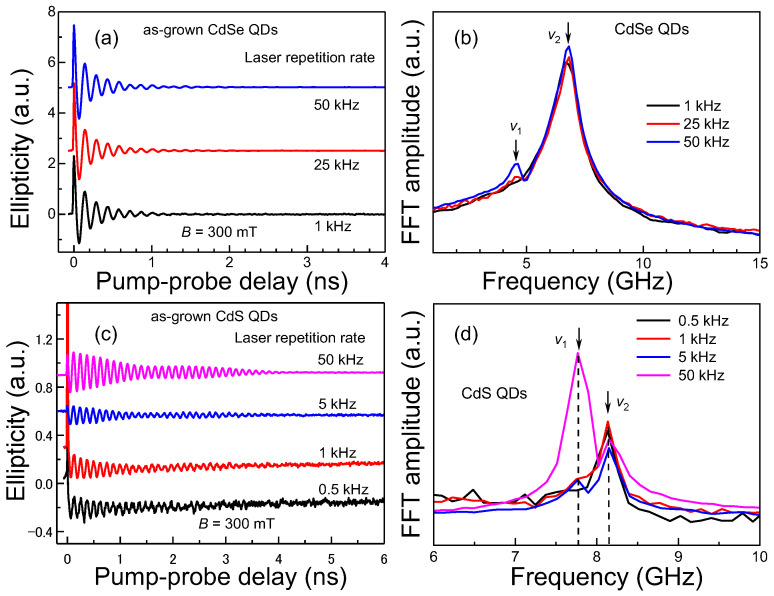
(**a**) Time-resolved ellipticity signals of CdSe QDs at laser repetition rates of 1, 25 and 50 kHz, respectively. (**b**) The corresponding FFT spectra of panel a. (**c**) Electron spin dynamics of CdS QDs with different laser repetition rates. (**d**) The corresponding FFT spectra of panel c.

## Data Availability

The data that support the findings of this study are available from the corresponding author upon reasonable request.

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
