# Peer review of "Methods for Obtaining One Single Larmor Frequency, Either v1 or v2, in the Coherent Spin Dynamics of Colloidal Quantum Dots"

_nanomaterials, 2023, doi:10.3390/nano13132006_

Round 1

Reviewer 1 Report

This paper reports interesting and important results on spin-dependent properties of CdSe and CdS colloidal nanocrystals. Namely, it demonstrates various approaches how to adjust the experimental conditions or technologically modify samples in order to give a favor in one of the two Larmor frequencies. Paper is clearly written and I recommend it for publication in Nanomaterials after considering following recommendations:

1. It would be very helpful, also in view of comparison of the presented results with the literature data, to give in text values of the Larmor frequencies and values of corresponding g-factors. The later is important as data for different magnetic fields are given.

2. The authors demonstrate various approaches to adjust the intensity of signals with different Larmor frequencies. But I am missing in paper clear identification to what physical processes (means also to what charging state of NC) these frequencies can be assigned. There is only one comment in Introduction with reference to [11]. It this paper more data are presented. Are they support the assignment made in Ref. [11], or there are some misfits with it? I recommend to include comments on that in summary part and conclusions.

3. Recent paper of Qiang et al, ACS nano 16, 18838 (2022) reports on spin coherence in CdSe NCs in glass. It has discussions on the two Larmor frequencies and their interpretation. Here also the theory of spin beats in NCs is revised and it is shown that in Ref. [5] exciton Larmor frequency was given not correct. I recommend to consider this paper in view of the reported results.

Reviewer 2 Report

The authors use time-resolved Faraday rotation measurement to investigate the spin dynamics in the CdS and CdSe quantum dots under different sample preparation and experimental conditions. They experimentally show that the double Larmor frequency can be driven to a single Larmor frequency by either (1) manipulating the hole acceptor prepared under different ambient conditions or/and (2) manipulating the repetition rate of the optical excitation. The experimental findings are explained by considering the location of the hole acceptor (i.e., it is at the core or surface of the QD) and the different charged state lifetime.

Overall, the results are sound and interesting. The draft was written quite clearly. The work is a significant enough extension of the previous work (ref [11]). Therefore, I recommend publication in Nanomaterials after minor revisions. Below are some comments that could improve the draft for the authors’ consideration.

1. In the abstract, there is a typo of “Lamor” instead of “Larmor”.

2. Is the pulse duration of 180 fs remain constant in all measurements? Can the author briefly discuss the effect of different pulse duration?

3. Based on the text, I understood that the samples with hole acceptor were prepared under different ambient conditions and, thus, have different behaviour. However, the ambient condition during the measurement is not clear. Is it always measured under ambient room atmosphere, vacuum, or pressurized N2, or is it varied from one measurement to another?

4.  It is better to include “CdS” and “CdSe” remarks inside the figure, especially in Fig. 4 and 5.

5. The authors explain that the effect of different laser repetition rates is mainly due to the different lifetimes of the charged states. What is the reason behind this difference in lifetime? Is this related to the location of the hole acceptor, i.e., whether the hole acceptor is on the surface or in the core?

6. The authors remark that Li[Et3BH] prepared under the N2 ambience condition will result in the v1 component and suppress the v2 component. How about Li[Et3BH] prepared under air atmosphere condition? Does this hole acceptor behave the same way as the OT acceptor, i.e., it will result in the v2 component if it is prepared under air atmosphere condition? If that is the case, is it possible that the main factor is the ambience condition during the preparation instead of the hole acceptor type?

Quality of English is sufficient.
